# ^68^Ga-Radiolabeling and Pharmacological Characterization of a Kit-Based Formulation of the Gastrin-Releasing Peptide Receptor (GRP-R) Antagonist RM2 for Convenient Preparation of [^68^Ga]Ga-RM2

**DOI:** 10.3390/pharmaceutics13081160

**Published:** 2021-07-28

**Authors:** Adrien Chastel, Delphine Vimont, Stephane Claverol, Marion Zerna, Sacha Bodin, Mathias Berndt, Stéphane Chaignepain, Elif Hindié, Clément Morgat

**Affiliations:** 1INCIA, University of Bordeaux, CNRS, EPHE, UMR 5287, F-33000 Bordeaux, France; adrien.chastel@chru-strasbourg.fr (A.C.); delphinevimont@wanadoo.fr (D.V.); sacha.bodin@chu-bordeaux.fr (S.B.); elif.hindie@chu-bordeaux.fr (E.H.); 2Nuclear Medicine Department, University Hospital of Bordeaux, F-33000 Bordeaux, France; 3Proteome Platform, University Bordeaux, F-33000 Bordeaux, France; stephane.claverol@u-bordeaux.fr (S.C.); stephane.chaignepain@u-bordeaux.fr (S.C.); 4Life Molecular Imaging (Formely Piramal Imaging) GmbH, 13353 Berlin, Germany; m.zerna@life-mi.com (M.Z.); m.berndt@life-mi.com (M.B.)

**Keywords:** GRP-R, ^68^Ga, PET, kit

## Abstract

Background: [^68^Ga]Ga-RM2 is a potent Gastrin-Releasing Peptide-receptor (GRP-R) antagonist for imaging prostate cancer and breast cancer, currently under clinical evaluation in several specialized centers around the world. Targeted radionuclide therapy of GRP-R-expressing tumors is also being investigated. We here report the characteristics of a kit-based formulation of RM2 that should ease the development of GRP-R imaging and make it available to more institutions and patients. Methods: Stability of the investigated kits over one year was determined using LC/MS/MS and UV-HPLC. Direct ^68^Ga-radiolabeling was optimized with respect to buffer (pH), temperature, reaction time and shaking time. Conventionally prepared [^68^Ga]Ga-RM2 using an automated synthesizer was used as a comparator. Finally, the [^68^Ga]Ga-RM2 product was assessed with regards to hydrophilicity, affinity, internalization, membrane bound fraction, calcium mobilization assay and efflux, which is a valuable addition to the in vivo literature. Results: The kit-based formulation, kept between 2 °C and 8 °C, was stable for over one year. Using acetate buffer pH 3.0 in 2.5–5.1 mL total volume, heating at 100 °C during 10 min and cooling down for 5 min, the [^68^Ga]Ga-RM2 produced by kit complies with the requirements of the European Pharmacopoeia. Compared with the module production route, the [^68^Ga]Ga-RM2 produced by kit was faster, displayed higher yields, higher volumetric activity and was devoid of ethanol. In in vitro evaluations, the [^68^Ga]Ga-RM2 displayed sub-nanomolar affinity (K*_d_* = 0.25 ± 0.19 nM), receptor specific and time dependent membrane-bound fraction of 42.0 ± 5.1% at 60 min and GRP-R mediated internalization of 24.4 ± 4.3% at 30 min. The [^nat^Ga]Ga-RM2 was ineffective in stimulating intracellular calcium mobilization. Finally, the efflux of the internalized activity was 64.3 ± 6.5% at 5 min. Conclusion: The kit-based formulation of RM2 is suitable to disseminate GRP-R imaging and therapy to distant hospitals without complex radiochemistry equipment.

## 1. Introduction

The Gastrin Releasing Peptide-Receptor (GRP-R) is a G-protein coupled receptor of the bombesin receptor family that show great potential for imaging and therapy of many cancers [1]. GRP-R is particularly overexpressed in prostate cancer, with low or no expression in healthy tissues. Its expression in prostate cancer is higher in carcinomas of lower Gleason score, lower tumor size and lower prostate specific antigen (PSA) value [2,3]. In pre-clinical studies and in pilot studies in humans, GRP-R imaging appears complimentary to that of imaging prostate-specific membrane antigen (PSMA) for patients with primary localized prostate cancer [4,5]. In recurrent prostate cancer, imaging the GRP-R may be beneficial in patients with negative or inconclusive [^18^F]Fluoroethylcholine imaging [6] and trials are undergoing to compare its role over [^68^Ga]Ga-PSMA-11 [7]. The initial steps of GRP-R-based targeted radionuclide therapy have been initiated recently [8].

Studies using immunohistochemistry or autoradiography on breast cancer samples carried out by us [9,10] and others [11] found GRP-R over-expression in ~70% of primary invasive breast cancers associated with the positivity of the estrogen receptor. GRP-R expression was also high in regional metastases from GRP-R positive primaries, raising hopes for radiopharmaceutical therapy of metastatic luminal breast cancer patients. Initial insight in patients support this finding [12]. Other cancer types might also benefit from GRP-R based imaging and therapy [1,13,14].

One key point to disseminate GRP-R imaging is the availability of kit-based formulation, thus avoiding the need for complex radiosyntheses and formulations using automated synthesizers [6,7,15]. Today, Positron Emission Tomography/Computed Tomography (PET/CT) of neuroendocrine tumors expressing somatostatin receptors is routinely performed with a kit-based formulation of DOTATOC (in Europe, SOMAKIT TOC) or DOTATATE (in the USA, NETSPOT) [16]. Similar achievements are expected with PSMA targeting in prostate cancer patients with PSMA-11 [17,18].

The main objective of this work was to validate a kit-based formulation of the GRP-R-antagonist RM2 for ^68^Ga radiolabeling and compare the process to that of a synthesizer-based method [10,15]. Importantly, [^68^Ga]Ga-RM2 is under evaluation in humans but it has not been fully characterized in vitro, and only microPET imaging of [^68^Ga]Ga-RM2 has already been described in an abstract [19]. A secondary objective was then to perform the first study characterizing the behavior in vitro of [^68^Ga]Ga-RM2 on PC-3 cells.

## 2. Materials and Methods

### 2.1. Description of RM2 Kits

Lyophilized kits containing 50 µg of RM2 acetate salt, 5 mg ascorbic acid (scavenger) and 50 mg trehalose (bulking agent) were provided by Life Molecular Imaging (Berlin, Germany). Figure 1 below represents the structure of the RM2 peptide.

### 2.2. Stability of the RM2 Kits

Kits were stored between 2 and 8 °C with automated temperature monitoring over a 12-month period. At selected time points (every 2 months), kits (in triplicate) were reconstituted in 5 mL ultra-pure water (VWR International, Fontenay-sous-Bois, France, Hypersolv Chromanorm for HPLC 83645.320) and subjected to UV-HPLC analysis (Phenomenex, Le Pecq, France, Luna C18; 250 mm × 4.6 mm × 5 µm; 0–1 min 95/5/1–7 min 5/95/7–8 min 5/95/8–9 min 95/5/9–10 min 95/5; 2.5 mL/min; H_2_O 0.1% TFA/Acetonitrile; 10 min) and liquid chromatography/mass spectrometry/mass spectrometry (LC/MS/MS) analysis (in singulet) (Lumos^®^, ThermoFischer Scientific, Waltham, MA, USA, injection volume 10 µL, pre-column 300 µm internal diameter × 5 mm C18 PepMapTM; column 75 µm internal diameter x 50 cm nanoViper C_18_, 2 µm, 100 Å—Acclaim^®^ PepMap RSLC; solvent A 95/05/0.1 H_2_O/ACN/HCOOH; solvent B 20/80/0.1 H_2_O/ACN/HCOOH; gradient 4–40% B in 50 min). Raw data were analyzed with BiopharmaFinder software (ThermoFischer Scientific, Waltham, MA, USA) allowing deconvolution of MS spectra during LC analysis. Fragments <1% were not considered. The sum of all significant fragments was set at 100%

### 2.3. Optimization of ^68^Ga Radiolabeling

Radiolabeling of RM2 kits was carried out using ^68^Ga chloride obtained from the marketed ^68^Ge/^68^Ga generator GalliAd^®^ (IRE Elit, Fleurus, Belgium, nominal activity of 1850 MBq, activity used in this work 350–900 MBq) in 1.1 mL HCl (already provided within the generator). Briefly, anhydrous acetate buffer (Emsure ACS, Merck, Darmstadt, Germany, Reag. Ph Eur ref AM1291668923) was dissolved in sterile and ready-to-use HCl 0.1 M (Rotem, KT720P). The requested volume was added in the kit up to the total targeted reaction volume (2.5 mL, 3.1 mL or 5.1 mL, corresponding to buffer volume of 1.4 mL, 2 mL and 4 mL, respectively, which are easy to prepare in radiopharmacies) and the generator is next eluted using a sterile, 25-mL vacuum vial. The raw solution was then heated (100 °C or 120 °C) using a heating block under continuous agitation for 8 or 10 min. The solution was then cooled at room temperature for 5 min, and 1.9 mL of PBS and 2 mL of 0.9% NaCl were added. Finally, the product was filtered using a sterilizing filter (Merck, Darmstadt, Germany, Millex-GV, SLGV013SL, 13 mm, 0.22 µm, PVDF). Influence of 4 h pre-elution of the generator was also tested.

## 3. Process Validation

### 3.1. Quality Control

For process validation, the [^68^Ga]Ga-RM2 kits were subjected to full quality control. Radiochemical purity (RCP) of [^68^Ga]Ga-RM2 was analyzed using UV-radio-HPLC ([^nat^Ga]Ga-RM2 was used to confirm the chemical identity) as described earlier and using thin layer chromatography (TLC) with ITLC-SG as stationary phase and methanol/ammonium acetate (50:50; *v*/*v*) as mobile phase. Molar activity was calculated after linear regression of the calibration curve with [^nat^Ga]Ga-RM2 (Life Molecular Imaging). The pH was determined using a pH-meter. Volumetric activity was determined by dividing activity obtained by the final volume. The [^68^Ga]Ga-RM2 was also subjected to endotoxin testing (Charles Rivers), sterility testing and radionuclidic purity and filter integrity tests. Results were then compared to the production capacity provided by the module route currently running in our NCT03604757 and NCT03606837 studies.

### 3.2. Costs

Costs needed for the kit production and the module route were compared. Only the direct costs were considered and encompassed reagent, small material for production and quality controls needed for each radiolabeling. Indirect costs such as the ^68^Ga generator, module, HPLC, TLC scanner, RM2 precursor (for module or kit), etc., were not included to provide a direct comparison.

## 4. In Vitro Characterization of [^68^Ga]Ga-RM2 Obtained from Kits

### 4.1. Lipophilicity

The lipophilicity of the [^68^Ga]Ga-RM2 was assessed by the water-octanol partition/distribution coefficient method. In a centrifuge tube 500 µL of 1-octanol was added to 500 µL of phosphate buffered saline (pH 7.4) containing the radiolabeled peptide (3.7 MBq; ≈10 µL). After equilibrium, the solution was vigorously stirred for 5 min at room temperature and subsequently centrifuged (4000 rpm, 3 min) to yield two immiscible layers. Aliquots of 100 µL were taken from each layer and the radioactivity in the samples was determined by a gamma counter (Gamma Wizard2 Counter 2480; Perkin Elmer, Villebon sur Yvette, France; acquisition time of one minute, energy window centered on the peak of 511 keV). The Log D_7.4_ was determined by the Log ratio (cpm (counts per minute) organic phase/cpm aqueous phase). The cpm values were decay-corrected. Experiments were performed three times in triplicates.

### 4.2. Cell Culture

For in vitro experiments, PC-3 cells were cultured in DMEM/F12 and seeded in 6-well plates at a density of 10^6^ cells per well 18 h before carrying out the experiments and incubated overnight with complete medium.

### 4.3. Saturation Radioligand Binding Assay

The affinity of the [^68^Ga]Ga-RM2 was studied on PC-3 cells. Well plates were first set on ice 30 min before the beginning of the experiment. [^68^Ga]Ga-RM2 was then added to the medium at concentrations of 0.1, 1, 10, 25 and 50 nM and cells were incubated (in triplicates) for 2 h at 4 °C. Incubation was stopped by removing the medium and washing the cells twice with ice-cold PBS. Finally, cells were treated with NaOH (1 M) and the radioactivity was measured in a gamma counter. In order to assess for non-specific affinity, excess non-radioactive bombesin (final concentration 1 μM), was added to selected wells. The specific binding was determined by subtracting the non-specific binding from the total binding. Dissociation constant K_d_ was then calculated using a one-site-specific binding model using GraphPad Prism^®^ software (V6.01 San Diego, CA, USA). Experiments were performed three times in triplicate. Results are expressed in specific binding (cpm) as a function of amount of radioligand added (nM) and expressed as mean ± standard deviation.

### 4.4. Determination of Specific Cellular Internalization and Membrane-Bound Fraction of [^68^Ga]Ga-RM2

The [^68^Ga]Ga-RM2 (1MBq, 165.6 ± 21.5 pmol, 32.9 ± 15.9 µL) was added to the medium and cells were incubated at 37 °C. At selected times (10, 30 and 60 min), internalization was stopped by removing medium and washing cells three times with ice-cold PBS. To determine the membrane-bound fraction, an acid wash was carried out twice using acetate buffer (pH = 5; 20 mM) at 4 °C for 5 min. Finally, cells were treated with NaOH (1 M) and the radioactivity was measured using a gamma counter. To determine non-specific internalization, 1 μM of bombesin was added to selected wells. Experiments were performed three times in triplicate. Results are expressed in percentage of specific internalized or membrane-bound fraction according to total cell-associated radioactivity and presented as mean ± standard deviation.

### 4.5. Determination of Cellular Efflux of [^68^Ga]Ga-RM2

The [^68^Ga]Ga-RM2 (1MBq, 499.3 ± 73.5 pmol, 99.9 ± 49.6 µL) was added to the complete medium and cells were incubated for 30 min at 37 °C. Three minutes before the end of the incubation time, internalization was stopped by removing medium and washing cells with ice-cold PBS. The membrane-bound fraction was removed by rinsing cells with 2 mL acetate buffer (pH = 5; 20 mM) for 2 min and cells were then rinsed ice-cold PBS. Next, 2 mL of complete medium was then added to each well, and PC-3 cells were incubated at 37 °C for 5, 15, 30 and 45 min. At each time point, efflux was stopped by removing medium and cells were rinsed twice with ice-cold PBS. Finally, the cells were treated with 1M NaOH and the radioactivity was measured using a gamma counter.

### 4.6. Calcium Mobilization Assay

For calcium imaging assays, PC-3 cells were cultured as previously described. The day before the experimentation, cells were grown on 15 mm Thermanox^®^ glass slides (Thermofischer, Waltham, MA, USA,) in 6-wells plates at the density of 5.10^5^ cells in 500 µL culture medium. Glass slides were individually loaded for 30 min in dark into 500 µL loading solution containing 10 µL fluo-8AM, 10 µL pluronic acid, 50 µL probenecid and 430 µL DMEM/F12 medium. Slides were then mounted on the imaging platform and perfused with buffer solution (containing (in mM): 130 NaCl, 3 KCl, 2.5 CaCl_2_, 1.3 MgSO_4_, 0.58 NaH_2_PO_4_, 25 NaHCO_3_ and 10 glucose) equilibrated with 95% O_2_/5% CO_2_, adjusted to pH 7.4 at room temperature (24–26 °C). A 10-min delay was applied to remove extracellular fluo-8AM and to enable fluo-8AM intracellular hydrolysis. Growing concentrations (10^−11^ M to 10^−5^ M) of [^nat^Ga]Ga-RM2 were added via a rapid perfusion system. Control experiments were performed by using bombesin (10^−11^ M to 10^−5^ M) and ionomycin (1 µM).

Imaging of intracellular Ca^2+^ changes was performed following excitation of cells at 490 nm with analysis of fluorescence emission at 514 nm using an Infinity 3 detector (Lumenera, Ottawa, ON, Canada) coupled with Eclipse E600FN microscope (Nikon, Tokyo, Japan). Data are expressed with F = medium basal fluorescence (at 514 nm) recorded before injection of the peptide, and ΔF is the difference between the peak value reached after peptide injection and the basal value.

### 4.7. Statistical Analysis

Data were compared using non-parametric *t* test (Wilcoxon test). Statistical analyses were performed using GraphPad software (v6.01, San Diego, CA, USA). *p* values < 0.05 were considered statistically significant.

## 5. Results

### 5.1. Stability of RM2 Kits

The quantitative study of RM2 kits by UV-HPLC over twelve months for storage at 2 °C to 8 °C found no significant differences in the RM2 amount over time (Figure 2A). The RM2 kits were also subjected to LC/MS/MS analysis, which allow qualification and quantification of the chemical species present in the kit over time. Three species were found in the kits, a 1638.8 Da dominant species corresponding to intact RM2 (Figure 2B), a 1621.8 Da fragment and an 801.4 Da fragment. Figure 2B shows the evolution of these three peptide fragments over time.

### 5.2. Optimization of RM2 Kits Radiolabeling with ^68^Ga

#### 5.2.1. Influence of Reaction Volume

In 2.5 mL volume the RCP value of 96.3 ± 0.1% was reached while in a volume of 3.1 mL, and the mean RCP was 97.1 ± 1.5%. Finally, in 5.1 mL, the mean RCP was 96.6 ± 2.3%. The pH in the reaction mixture was 3.0 for all experiments.

#### 5.2.2. Influence of Heating Time

In 3.1 mL, heating for 8 min provides a mean RCP of 88.9 ± 4.4% (Figure 3A). An additional 2 min of heating (total time of 10 min) allows reaching a mean RCP of 96.9 ± 1.6% (Figure 3B).

#### 5.2.3. Influence of Temperature

Two temperature conditions were applied for a fixed time of incubation of 10 min as determined above. At 100 °C, the mean RCP was 97.0 ± 1.6%, while at 120 °C, the RCP was 98. 5 ± 1.1%.

#### 5.2.4. Influence of Previous 4-h Generator Elution

Pre-elution of the generator 4 h before specific elution for radiolabeling experiment leads to a mean RCP of 96.94 ± 1.62%. In absence of this pre-elution step, the mean RCP was 91.50 ± 3.54%.

#### 5.2.5. Quality Controls of [^68^Ga]Ga-RM2

After optimization, ^68^Ga radiolabeling of the RM2 kits was carried out in three consecutive batches, and they all complied with the requirements of the monograph of the European Pharmacopoeia (as described for [^68^Ga]Ga-edotreotide) (Figure 4 and Table 1). The evolution of the radiochemical purity over time was also checked. The RM2 kits radiolabeled with ^68^Ga were found to be stable in vehicle up to 3 h.

## 6. Comparison of the Kit Formulation to the Module Production Route

In this study, we were also able to compare different production routes for [^68^Ga]Ga-RM2 (kit-based formulation *vs.* synthesis module). In both cases, the radiochemical purity of the obtained product complies with the requirements of the European Pharmacopoeia, with, however, better radiochemical purity was reached using the module route (99.4 ± 0.4 vs. 96.7 ± 1.4%; *p* < 0.0001) due to the additional purification step using C_18_ sep-pack. A lower molar activity was reached using the kit production given the higher amount of precursor engaged in the radiolabeling reaction (7.2 ± 1.3 vs. 13.2 ± 6.4 GBq/µmol; *p* = 0.03). This drawback is offset by the higher volumetric activity (*p* = 0.03) and the higher radiolabeling yield obtained via the kit route (81.1 ± 1.1% vs. 48.9 ± 12.9% for the module; *p* < 0.0001, non-decay-corrected). Importantly, the production of the radiolabeled molecule by kit was found to be less time-consuming than with the module (~20 min vs. ~40 min) as well as less expensive (EUR 65.7 vs. EUR 282.1) (Table 2). Indeed, these costs are site-specific and need to be adapted accordingly, but they provided an overall estimation.

## 7. Characterization of [^68^Ga]Ga-RM2 Produced by Kits

### 7.1. Determination of Lipophilicity

By using the partition method, a LogD (pH 7.4) value of −2.54 ± 0.04 was found for [^68^Ga]Ga-RM2, indicating a high hydrophilicity.

### 7.2. Saturation Radioligand Binding Assay

The specific receptor binding of [^68^Ga]Ga-RM2 for GRP-R was investigated on PC-3 cells. Saturation binding curves revealed a sub-nanomolar affinity was with a K*_d_* value of 0.25 ± 0.19 nM (Figure 5A).

### 7.3. Determination of Cellular Internalization and Membrane-Bound Fraction of [^68^Ga]Ga-RM2

The GRP-R mediated internalization and the GRP-R membrane-bound fraction of [^68^Ga]Ga-RM2 into PC-3 cells were analyzed. Specific and time-dependent internalization into PC-3 cells was observed with a maximum of 24.4 ± 4.3% of the cell-associated radioactivity being internalized at 30 min (corresponding to 0.8–5.6 fM/mg protein). In addition, a receptor specific and time dependent membrane-bound fraction of [^68^Ga]Ga-RM2 was seen with a maximum of 42.0 ± 5.1% at 60 min (Figure 5B) corresponding to 2.1–10.3 fM/mg protein.

### 7.4. Determination of Cellular Efflux of [^68^Ga]Ga-RM2

The [^68^Ga]Ga-RM2 was further evaluated regarding cellular efflux on PC-3 cells. A high and fast efflux of internalized radioactivity was found for [^68^Ga]Ga-RM2. Already 5 min post-internalization, 64.3 ± 6.5% of the total binding was externalized. The efflux was roughly constant over time up to 45 min (~65–70%) (Figure 5C).

### 7.5. Intracellular Calcium Mobilization Assay

The calcium release was evaluated after the application of increasing concentrations of bombesin or [^nat^Ga]Ga-RM2 on PC-3 cells. Bombesin effectively induced calcium mobilization in a dose-dependent manner while [^nat^Ga]Ga-RM2 completely abolished calcium release (Figure 6). Ionomycin at 1 µM was used as additional positive control.

## 8. Discussion

The availability of radiopharmaceuticals for molecular imaging is critical to improve patient care. GRP-R has the potential to improve management of patients suffering from various cancers, with highly promising applications in prostate cancer and breast cancer [13]. To disseminate radiopharmaceuticals for imaging and therapy of GRP-R, ready-to-use kits are prerequisites for the rapid and reproducible preparation of radiopharmaceuticals. In this work we have characterized a kit-based formulation of RM2, a potent GRP-R antagonist [20]. By using an innovative methodology combining LC/MS/MS and UV-HPLC, we were able to demonstrate that the RM2 peptide remains intact after 1 year of storage between 2 and 8 °C. Interestingly, two minor fragments (<5% each) of 1621.8 Da and 801.4 Da were found by LC/MS/MS (Figure 2). The finding of common amino terminal fragments between the 1621.8Da fragment and intact RM2 indicates that the loss of 17 Da was related to the carboxy terminal position of RM2. The 801.4 Da fragment is due to the loss of an 837 Da fragment from the parent RM2 and was found at very low amount (<2%). These impurities were found already at the beginning of the study and remained constant over time. Additional analyses, for example, during the synthesis of the RM2 precursor, are necessary to elucidate the origin of these impurities. The UV-HPLC study confirms the stability of the RM2 peptide but the 1621 Da impurity was not detected presumably due to a lower sensitivity than LC/MS/MS.

Next, we aimed at establishing a radiolabeling protocol with ^68^Ga fulfilling the requirements of the European Pharmacopoeia. We found that a pH 3.0 value leads to maximum radiolabeling yield. It is known that complexation of Ga^3+^ ion with the DOTA macrocycle is optimal at pH 3 to 4 [21]. Regarding the buffer, we choose to use acetate buffer rather than HEPES given that the latter is not approved for human use and requires an additional purification step and because acetate buffer is a pharmacologically harmless buffer [22]. The reaction volume, temperature and heating time were also optimized. We experimentally demonstrated that the ^68^Ga incorporation tolerated a total reaction volume (including generator) of 2.5–5.1 mL and heating at 100 °C for 10 min under gentle agitation. For convenient preparation, we choose to use the 3.1 mL volume (2 mL acetate buffer plus 1.1 mL from the generator). Moreover, heating is a compulsory step because the DOTA macrocycle needs to adopt a pseudo-octahedral configuration to allow gallium ion binding in a cis-coordinated complex [23]. It was also necessary to find a compromise between a heating time long enough to allow a sufficient labeling yield, but short enough to accommodate with the half-life of ^68^Ga (67.7 min) for its ultimate use in the daily clinical routine. After heating, a time for cooling down has been set up to allow radiolabeling to be completed with the residual thermal energy, before formulation of the [^68^Ga]Ga-RM2 kit compatible with intravenous application. The stability of [^68^Ga]Ga-RM2 preparation in the vehicle was also studied over 3 h and confirm the possibility of use in the daily hospital routine, while allowing flexibility in the nuclear medicine service in case of unexpected events which can interfere with the patient’s injection.

For instance, considering an elution of 1 GBq of ^68^Ga, the additional activity available due to the shorter production time and higher radiolabeling yield with the kit would be ~300 MBq which allow the injection of at least two additional patients. The kit also has the advantage of being devoid of ethanol, contrarily to the module route which requires ethanol during the purification process. The kit is also composed of trehalose for lyophilization (composition without trehalose have not met criteria for lyophilization). The module route to produce [^68^Ga]Ga-RM2 does not use trehalose meaning that trehalose in the kit does not interfere with ^68^Ga complexation. Finally, radiolysis and lower radiochemical purity (<80%) were obtained if radiolabeling were performed with kit compositions without ascorbic acid highlighting the role of scavenger (ascorbic acid).

Another kit formulation of RM2 has been developed [24] but without a scavenger and a bulking agent for lyophilization. Surprisingly, in this study the specific uptake of [^68^Ga]Ga-RM2 in PC-3 looks rather low (5% vs. 60% of cell-associated radioactivity in this study). Therefore, our innovative formulation provides longer stability at higher temperature and higher uptake in GRP-R-expressing PC-3 cells.

In recent years, targeting of GRP-R has been the subject of clinical studies, demonstrating growing interest of prostate cancer and breast cancer imaging. Surprisingly, [^68^Ga]Ga-RM2 has barely been studied in vitro and was directly investigated using µPET/CT on mice and PET/CT in humans. Therefore, we aimed at investigating the in vitro behavior of [^68^Ga]Ga-RM2, after kit-based production. As reported in Table 3, [^68^Ga]Ga-RM2 exhibited improved GRP-R-affinity compared to [^111^In]In-RM2, demonstrating that affinity should be carefully studied when changing the radionuclide as already described [25,26].

[^68^Ga]Ga-RM2 also showed a two- to threefold improvement in GRP-R-mediated internalization and membrane-bound fraction, in line with the higher affinity towards the GRP-R. Interestingly, the membrane-bound fraction/internalized fraction ratio of [^68^Ga]Ga-RM2 remains roughly similar to that of [^177^Lu]Lu-RM2 and [^111^In]In-RM2 (Table 3). The efflux of [^68^Ga]Ga-RM2 was found to be around 70%. Although this value looks high, we should keep in mind that only the internalized fraction is externalized and that this value does not consider the redistribution mechanism of the radiopharmaceutical. Given the higher signal found on the membrane, this value should not prevent obtaining PET images of high quality in the clinical setting. Finally, intracellular calcium mobilization assay was performed and confirmed that [^nat^Ga]Ga-RM2 still is an antagonist to the GRP-R.

## 9. Conclusions

In this work, we have developed a kit-based formulation of RM2 suitable for convenient ^68^Ga-radiolabeling in hospitals not equipped with an automated synthesizer. Moreover, the kit-based production was more efficient than the module production route. Finally, we demonstrated in vitro that [^68^Ga]Ga-RM2 displayed higher affinity and higher binding at the GRP-R when compared with the ^111^In and ^177^Lu counterparts.

## Figures and Tables

**Figure 1 pharmaceutics-13-01160-f001:**
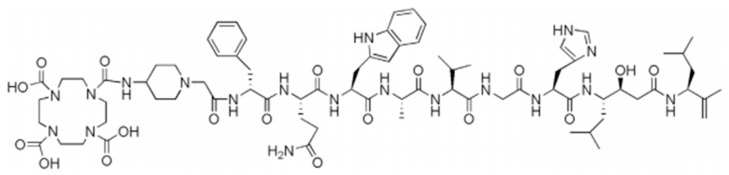
Structure of the RM2 peptide.

**Figure 2 pharmaceutics-13-01160-f002:**
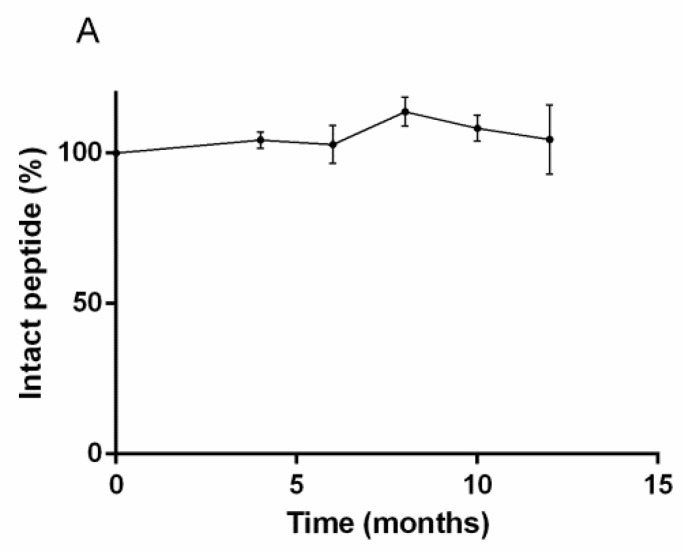
(**A**) UV-HPLC analysis of RM2 kit over time. Only one peak corresponding to intact RM2 was identified and remains stable over-time. (**B**) LC/MS/MS analysis of RM2 kit demonstrating the presence of intact RM2 and two fragments, the proportions of which remain constant over one year.

**Figure 3 pharmaceutics-13-01160-f003:**
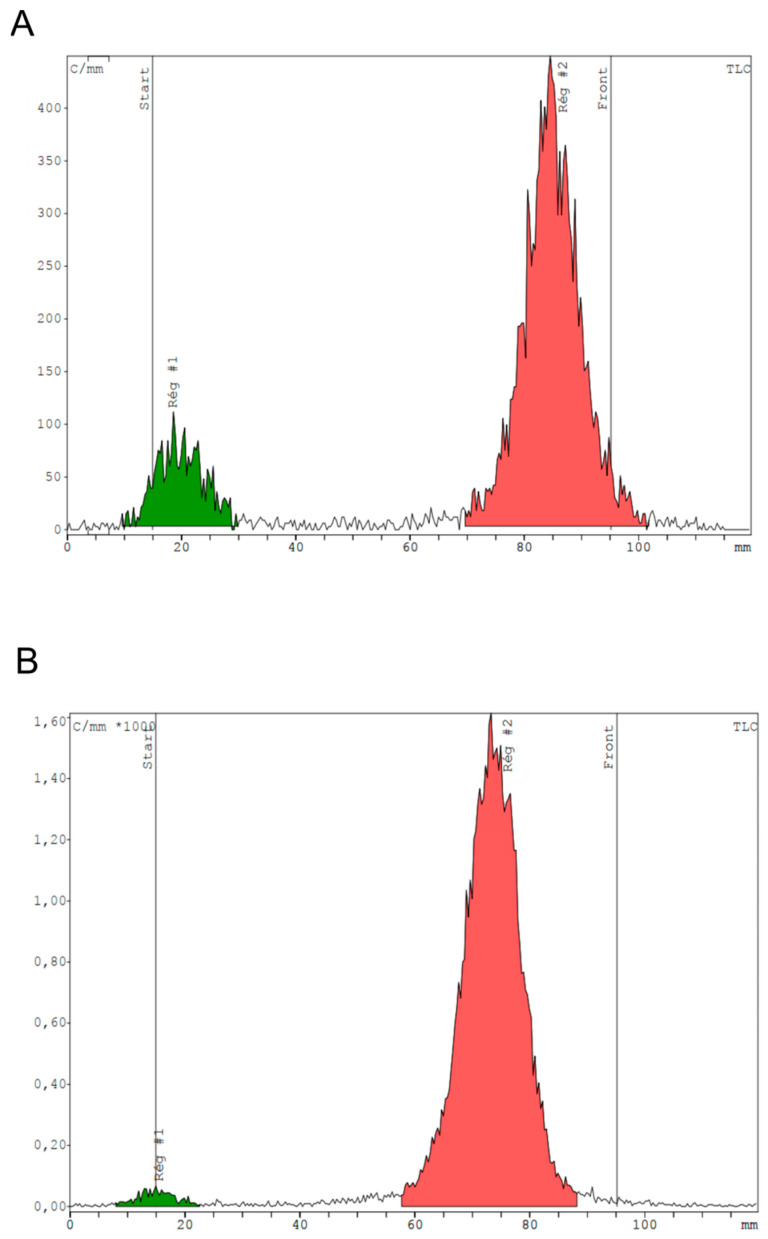
Representative TLC radiochromatograms showing radiochemical purity of [^68^Ga]Ga-RM2 obtained after 8 min heating (**A**) vs. 10 min (**B**) in a reactional volume of 3.1 mL. Reg #1 stands for region 1 and Reg#2 means region 2.

**Figure 4 pharmaceutics-13-01160-f004:**
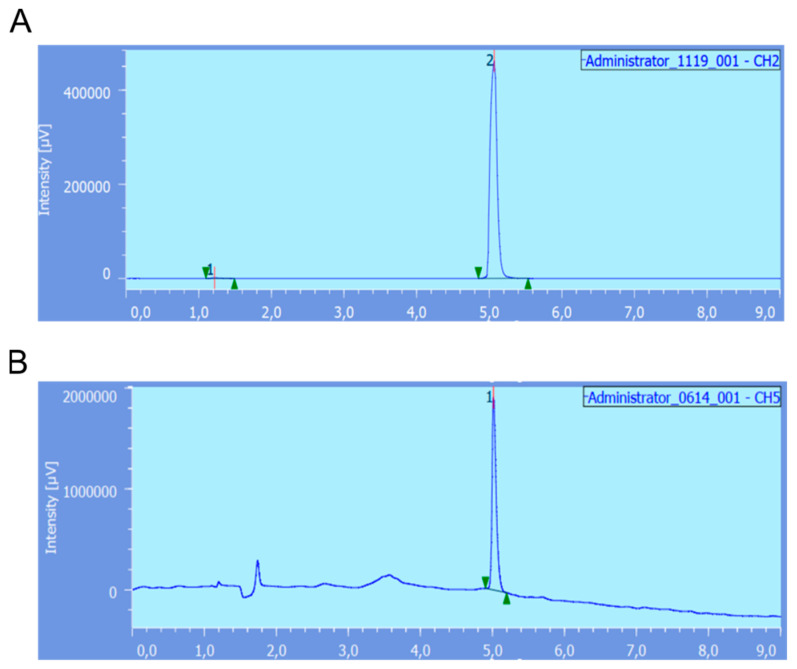
(**A**) radio-HPLC trace of [^68^Ga]Ga-RM2 obtained after direct radiolabeling of the kit developed in this study. Chemical identity was checked using [^nat^Ga]Ga-RM2 (UV-HPLC trace (**B**)).

**Figure 5 pharmaceutics-13-01160-f005:**
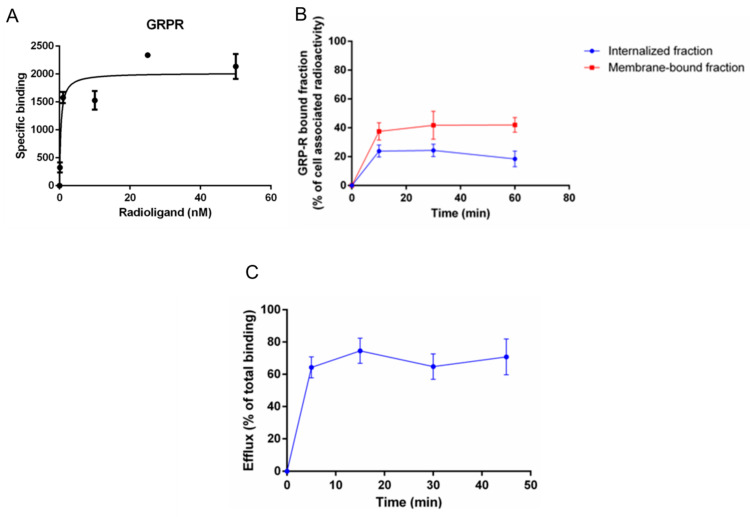
Saturation binding curve of [^68^Ga]Ga-RM2 towards the GRP-R (**A**), internalized and membrane-bound fractions of [^68^Ga]Ga-RM2 (**B**) and total efflux of [^68^Ga]Ga-RM2 (**C**). All experiments were performed in triplicates on the PC-3 cell line.

**Figure 6 pharmaceutics-13-01160-f006:**
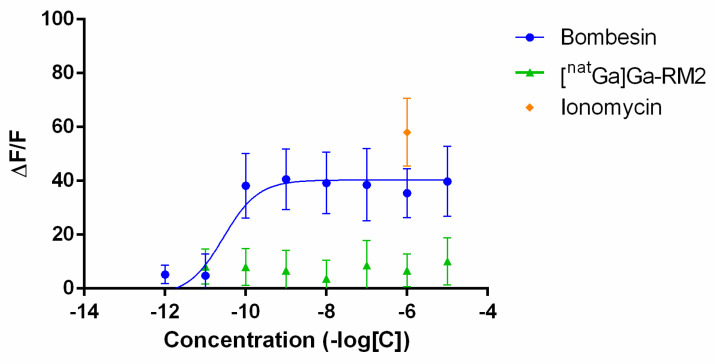
Intracellular calcium release induced in PC-3 cells by bombesin, [^nat^Ga]Ga-RM2 and ionomycin. Results are shown as the difference between the peak value reached after peptide injection and the basal value divided by medium basal fluorescence (ΔF/F).

**Table 1 pharmaceutics-13-01160-t001:** Validation batches of [^68^Ga]Ga-RM2 obtained from the kit developed in this study.

Test	Specifications	Batch 1	Batch 2	Batch 3
Volumetric activity (MBq/mL)	na	22.65	28.00	20.23
Molar activity (GBq/µmol)	na	8.29	5.8	7.6
Appearance	Clear and colourless	Complies	Complies	Complies
pH	4–8	4.7	4.7	4.7
Identification of ^68^Ga	511/1022/1077 keV	Complies	Complies	Complies
	62–74 min	67.17	67.90	67.61
Radiochemical purity (%)	≥91 (HPLC)	97.5	98.6	95.1
	≥91 (TLC)	95.3	98.3	94.4
Amount of [^68^Ga]Ga-RM2 (µg)	<50	38.1	45.3	31.8
Bubble point test	≥3.4 bars	3.6	3.6	3.7
Bacterial endotoxins (EU/mL)	<17.5	<5	<5	<5
Sterility	Sterile	Complies	Complies	Complies

**Table 2 pharmaceutics-13-01160-t002:** Comparison of direct costs for the production of [^68^Ga]Ga-RM2 by module route or by kit.

		Cost per Synthesis (€) MODULE	Cost per Synthesis (€) KIT
Chemicals	Water for chromatography	5.8	
	Ethanol	0.18	
	Sodium acetate	0.22	0.003
	Hydrochloric acid		0.83
	Sodium hydrogenophophate	0.002	0.002
	Sodium chloride	0.14	0.0001
	NaCl 0.9%	0.72	0.72
	Potassium dihydrogenophosphate	0.0006	0.0006
	Trifluoro acetic acid	0.99	0.99
Consumables	Module cassette (including tubes, etc…)	197.5	
	C_18_ sep-pack	4.24	
	Needles	0.28	0.66
	Syringes	1.05	0.48
	Pipette tips	0.81	
	Gloves	0.39	0.39
	Vials	16.14	8.07
	Filters	1.94	1.94
	Endotoxins cartridges	51.6	51.6
Total		282.1	65.7

**Table 3 pharmaceutics-13-01160-t003:** Comparison of in vitro radiopharmaceutical properties of [^68^Ga]Ga-RM2 over [^111^In]In-RM2 [20] and [^177^Lu]Lu-RM2 [27].

	[^68^Ga]Ga-RM2	[^111^In]In-RM2	[^177^Lu]Lu-RM2
LogD_7.4_	−2.54 ± 0.04	nd	nd
Affinity (Kd in nM)	0.25 ± 0.19	2.9 ± 0.4	nd
Membrane bound fraction (%)	42.0 ± 5.1 (1 h)	15.9 ± 0.9 (4 h)	11.2 ± 0.8 (4 h)
Internalization (%)	18.5 ± 5.4 (1 h)	3.7 ± 0.4 (4 h)	4.5 ± 0.6 (4 h)
Efflux (%)	70.8 ± 11.1 (45 min)	nd	nd

nd: not determined.

## Data Availability

The datasets used and/or analyzed during the current study are available from the corresponding author on reasonable request.

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
