# Peer review of "68Ga-Radiolabeling and Pharmacological Characterization of a Kit-Based Formulation of the Gastrin-Releasing Peptide Receptor (GRP-R) Antagonist RM2 for Convenient Preparation of [68Ga]Ga-RM2"

_pharmaceutics, 2021, doi:10.3390/pharmaceutics13081160_

Round 1
Reviewer 1 Report
The authors present a straight-forward report for the development of a kit labelling method for the GRP targeting peptide RM2 with 68Ga. The conclusions are well supported by the data. While there is perhaps limited impact for a paper on this topic, the ability to develop kits for 68Ga labelling could be of importance for some radiopharmacy operations and may provide a means for reduced cost for producing metal-based PET agents.
Recommended revisions:
- Figures require improvement for presentation quality. For example, the HPLC trace in figure 2 has an awkward blue background with white text.
- Please include a figure showing the structure of the RM2 peptide. I realize this can be found through the references provided, but I think it is important that the reader has an idea of the type and size of the molecule being labelled.
- The authors should follow the accepted radiochemistry nomenclature, for example, specific activity should be replaced with molar activity. Please see the article, “Consensus nomenclature rules for radiopharmaceutical chemistry — Setting the record straight” https://doi.org/10.1016/j.nucmedbio.2017.09.004
- Significant figures being used for data at times appears too accurate. Are you really able to determine purity to +/- 0.01 % accuracy? Please review all data to insure significant figures are within reason for the techniques being used. Also, be consistent for indicating units within a table.
- Please review for readability, as there are occasional statements that are poorly worded or not well explained. Examples, “The raw solution was…” and “Generator was eluted 4h before radiolabeling or not.”
- The reader cannot determine the legitimacy of the cost comparison, whether the calculation is appropriate or whether it is a corporate sales pitch. Perhaps a supporting information section could be added for those who wish to see the methodology that was used (if this journal supports having a SI section). I suspect that this type of calculation can be easily swayed depending upon the parameters considered.
Author Response
Reviewer 1
The authors present a straight-forward report for the development of a kit labelling method for the GRP targeting peptide RM2 with 68Ga. The conclusions are well supported by the data. While there is perhaps limited impact for a paper on this topic, the ability to develop kits for 68Ga labelling could be of importance for some radiopharmacy operations and may provide a means for reduced cost for producing metal-based PET agents.
Recommended revisions:
- Figures require improvement for presentation quality. For example, the HPLC trace in figure 2 has an awkward blue background with white text. Thank you for your nice reading of our work. We have replaced the figure by a clean figure without the white text.
- Please include a figure showing the structure of the RM2 peptide. I realize this can be found through the references provided, but I think it is important that the reader has an idea of the type and size of the molecule being labelled. Thank you for your proposal. We have included the semi-developed structure of RM2 in a new figure.
- The authors should follow the accepted radiochemistry nomenclature, for example, specific activity should be replaced with molar activity. Please see the article, “Consensus nomenclature rules for radiopharmaceutical chemistry — Setting the record straight” https://doi.org/10.1016/j.nucmedbio.2017.09.004 . Thank you for your remark, we have modified accordingly.
- Significant figures being used for data at times appears too accurate. Are you really able to determine purity to +/- 0.01 % accuracy? Please review all data to insure significant figures are within reason for the techniques being used. Also, be consistent for indicating units within a table. Thank you for your remark. We have modified the purity values to +/- 0.1% which is in the acceptable range for radioactivity analysis. We have also checked the tables and added the missing units.
- Please review for readability, as there are occasional statements that are poorly worded or not well explained. Examples, “The raw solution was…” and “Generator was eluted 4h before radiolabeling or not.”. Thank you for your nice reading of our work which improves the quality of the manuscript. We have rephrase these statements
- The reader cannot determine the legitimacy of the cost comparison, whether the calculation is appropriate or whether it is a corporate sales pitch. Perhaps a supporting information section could be added for those who wish to see the methodology that was used (if this journal supports having a SI section). I suspect that this type of calculation can be easily swayed depending upon the parameters considered. Authors understand the point of view of the reviewer. For more clarity we have added a new table showing all the material used for radiolabeling, formulation and QC) considered for this calculation. We have also toned down our results regarding costs by indicating that these costs are site-specific. Thank you
Reviewer 2 Report
The manuscript entitled "68Ga-radiolabeling and pharmacological characterization of a kit-based formulation of the Gastrin-Releasing Peptide Receptor (GRP-R) antagonist RM2 for convenient preparation of [68Ga]Ga-RM2" submitted by Chastel A et al describes a kit based formulation of GRP-R antogonist RM2 with 68Ga radiolabeling. Please see comments below:
1) The abstract has typos, please correct and proofread them. Please describe or reference what RM2 is ?
2) For optimization of 68Ga radiolabeling, when does RM2 was added and how the labeling works? Does 68Ga coordinates with RM2? Optimization procedure was not clear. What does "generator was eluted 4h before radiolabeling or not" What does it tells?
3) How does the author confirmed the radiolabeled 68Ga-RM2 was obtained?
4) For specific cellular internalization, does the authors determine uptake of 68Ga-RM2 per mg or microgram of protein after decay and determine the protein concentration? Please provide pictures for internalization and clear procedure.
5) Please provide size exclusion column chromatography for RM2 at 0 min, 2 months and show SEC chromatograms for the manuscript?
6) In fig 1B, what are 1621.8 Da and 801.4 Da fragments refers to? Please analyze and discuss.
7) Please include SEC for 68Ga-RM2 at 100 oc and 120 oc and test their thermal stability?
8) For figure 2, please include uv trace and coinjection of cold RM2 for identification of RM2. If RM2 kit was a mixture of components mentioned in page 3, where does Ga68 binds?
9) For fig 3c, how did the authors determine extracellular activity?
10) For fig 4, please include ionomycin for more data points for comparison, single data point is not enough.
2)
Author Response
Reviewer 2
The manuscript entitled "68Ga-radiolabeling and pharmacological characterization of a kit-based formulation of the Gastrin-Releasing Peptide Receptor (GRP-R) antagonist RM2 for convenient preparation of [68Ga]Ga-RM2" submitted by Chastel A et al describes a kit based formulation of GRP-R antogonist RM2 with 68Ga radiolabeling. Please see comments below:
1) The abstract has typos, please correct and proofread them. Please describe or reference what RM2 is ? Thank you for your remark. We have added a new figure showing the structure of the RM2 peptide
2) For optimization of 68Ga radiolabeling, when does RM2 was added and how the labeling works? Does 68Ga coordinates with RM2? Optimization procedure was not clear. What does "generator was eluted 4h before radiolabeling or not" What does it tells? The methodology we used is very common for radiolabeling with radiometals (see our review Morgat et al, gallium-68: chemistry and radiolabeled peptides exploring different oncogenic pathways. 2013). The methodology used to radiolabel the kits is described in details in the manuscript. Briefly, the acetate buffer was first prepare and added to the kit. Next the generator is eluted in the kit dissolved in acetate buffer. Regarding the generator, authors would provide to the reviewer the following information. A gallium generator is composed of a mother radionuclide 68Ge (adsorbed into a metallic column) which decays by electronic capture in pure 68Ga. Physical half-life of 68Ge is 271 days vs 68 min for 68Ga. Finally 68Ga decays in 68Zn, a stable metal. Therefore, the gallium generator contains metals which can interfere with the radiolabeling and can be cleaned by an elution. This is the reason why we tested the influence of a 4h-pre elution and direct labelling without 4h pre-elution. 4h is the best compromise between 68Ga-activity available and accumulation of metal and is largely used in centers equipped with 68Ga. A comprehensive review by Velikyan et al 2015 explains in details all the rational of this approach. For more clarity we have rephrase according: “Influence of 4h pre-elution was also tested”. Thank you
3) How does the author confirmed the radiolabeled 68Ga-RM2 was obtained? Thank you for your remark. We have confirmed that 68Ga-RM2 was obtained by injected natGa-RM2 into HPLC. Both 68Ga-RM2 and natGa-RM2 have indeed the same retention time. This has been added to the manuscript: [natGa]Ga-RM2 was used to confirm the chemical identity
4) For specific cellular internalization, does the authors determine uptake of 68Ga-RM2 per mg or microgram of protein after decay and determine the protein concentration? Please provide pictures for internalization and clear procedure. The internalization is expressed in % of specific binding according to total binding. We do not provide uptake per milligram of protein because decay (> 10h) is needed before measuring protein concentration. However, PC-3 cells grow fast and the > 10h-waiting needed for decay would greatly modify the protein concentration. Therefore, we provide uptake expressed in % of specific according to total binding. The procedure for internalization is fully described in the paragraph “determination of specific cellular internalization and membrane bound fraction of [natGa]Ga-RM2. Figure 5B showed the intracellular behavior of [68Ga]Ga-RM2 on PC3 cells
5) Please provide size exclusion column chromatography for RM2 at 0 min, 2 months and show SEC chromatograms for the manuscript? Thank you for your proposal. SEC is not usual for radiopharmaceutical development. Moreover, in the seminal paper of Mansi and co-workers (Mansi et al Eur J Nucl Med Mol Imaging 2011), who developed RM2, authors have used UV-radioHPLC not SEC. Therefore, we are not very comfortable in using SEC for these reasons.
6) In fig 1B, what are 1621.8 Da and 801.4 Da fragments refers to? Please analyze and discuss. Thank you for your remark, we were also interested in knowing what these fragments are. The 1621.8Da results in the loss of 17Da fragment from the parent peptide RM2 (1638.8Da). We hypothesized that the loss of a water molecule would be an explanation. However, the loss of water would be 18Da and not 17Da, but the loss of hydroxy is also not correct because the position of hydroxy loss would have a hydrogen in its place (CO2H to CHO) and would be a loss of 16Da not 17Da. Loss of water is more likely than the reduction of an acid to aldehyde in peptide synthesis due to the use of dehydrating reagents used in the coupling steps. The only mass we could find was hydrolysis of the C-terminal amide to an acid which then forms a 7-membered ring with the hydroxyl of the Statine and then the mass fits. 17Da would also be the loss of ammonia NH3, which could indicate glutarimide formation. As we are not able to present experimental data supporting these hypothesis we suggest to state that no obvious ideas and further work would be needed to identify what these fragments are. This is the reason why we do not give more information within the manuscript. Thank you
7) Please include SEC for 68Ga-RM2 at 100 oc and 120 oc and test their thermal stability? Please see our comment regarding SEC at point 5.
8) For figure 2, please include uv trace and coinjection of cold RM2 for identification of RM2. If RM2 kit was a mixture of components mentioned in page 3, where does Ga68 binds? Thank you for your remark. Ga3+ ion is a metal which require coordination chemistry. Therefore, the only position where Ga3+ is bound is within the DOTA macrocycle (part of the RM2 peptide). As suggested we have modified the figure by adding the UV trace of [natGa]Ga-RM2
9) For fig 3c, how did the authors determine extracellular activity? Regarding this point, the methodology is fully described in the paragraph “determination of cellular efflux of [68Ga]Ga-RM2. Briefly, cells were incubated with the radiopharmaceutical and at selected time points, externalized radioactivity in the culture medium was collected and measured in a gamma counter.
10) For fig 4, please include ionomycin for more data points for comparison, single data point is not enough. Thank you for your remark. For this experiment we have followed the method described by Mansi et al Eur J Nucl Med mol Imaging 2010 in which a single point of ionomycin was used along with increasing concentration of bombesin. Thank you
Reviewer 3 Report
The research article " 68Ga-radiolabeling and pharmacological characterization of a kit-based formulation of the Gastrin-Releasing Peptide Receptor (GRP-R) antagonist RM2 for convenient preparation of [68Ga]Ga-RM2" authored by Chastel et al is a comprehensive study.
Comments;
The conclusion section needs more clarity.
Author Response
Reviewer 3
The research article " 68Ga-radiolabeling and pharmacological characterization of a kit-based formulation of the Gastrin-Releasing Peptide Receptor (GRP-R) antagonist RM2 for convenient preparation of [68Ga]Ga-RM2" authored by Chastel et al is a comprehensive study. Thank you for your positive comment of our work
Comments;
The conclusion section needs more clarity. Thank you for your comment. We have clarified the conclusion as follow: [Finally, we demonstrated in vitro that [68Ga]Ga-RM2 displayed higher affinity and higher binding at the GRP-R when compared with the 111In and 177Lu counterparts]
Reviewer 4 Report
This paper reported a practical method for 68Ga-radiolabeling of GRP-R. The experiments were properly designed and could support the authors' hypothesis. The paper is well written and easy to read. I recommend its publication in the journal of Pharmaceutics.
Author Response
Reviewer 4
This paper reported a practical method for 68Ga-radiolabeling of GRP-R. The experiments were properly designed and could support the authors' hypothesis. The paper is well written and easy to read. I recommend its publication in the journal of Pharmaceutics.
We thank the reviewer for his comment
Reviewer 5 Report
In this manuscript Authors present comprehensively described studies on new kit-based formulation for GRP-R antagonist (RM2) labelled with diagnostic 68Ga. The aim of this study is to evaluate the potential of application RM2 kit for imaging GRP-R overexpressed tumors, such as prostate and breast cancer. The Authors described both chemical and biological characteristics of synthesized radiobioconjugate with using pharmacopeia recommended techniques. Disadvantage of in vitro studies is using only one cell line (PC-3).
In general, manuscript is clearly described and contains all the relevant research elements in this field, along with their interpretation. Several major revisions may improve the manuscript before further processing and publication in Pharmaceutics.
- In section “Optimization of 68Ga radiolabeling” please specify what was 68Ga activity used for labeling.
- In section “Optimization of 68Ga radiolabeling” please explain why did you chose this specific volumes for testing (taking into account that 1.1 mL in each of them is 68Ga).
- In section “Determination of specific cellular internalization and membrane-bound fraction of [68Ga]Ga-RM2” you mentioned, that for membrane-bound fraction removal, acetate buffer with pH 5.0 was used. Usually, for this step, glycine or any other buffer with pH around 2.5-3.0 due to its more acidic properties. I am not sure, that quite significant difference in pH has no influence for effectiveness of membrane-bound conjugates removal. Please comment.
- In section RESULTS “Optimization of RM2 kits radiolabeling with 68Ga” please add iTLC and/or HPLC confirmation of RCP results.
- In section “Comparison of the kit formulation to the module production route” please detail what is additional purification step in module synthesis.
- Please make spaces between digit and unit (e.g. Materials and Methods; Description of RM2 kits, 5mg, etc.)
Summary:
Studies described in this manuscript were performed, described and interpreted carefully and all of the results were precisely discussed. Of course, using only one cell line is not the most appropriate in case of in vitro studies and might be improved. Worth attention is fact, that Authors followed European Pharmacopeia criteria for evaluation of radiobioconjugate application potential. After adding iTLC or HPLC confirmation of RCP results manuscript will be significantly improved.

Author Response
Reviewer 5
In this manuscript Authors present comprehensively described studies on new kit-based formulation for GRP-R antagonist (RM2) labelled with diagnostic 68Ga. The aim of this study is to evaluate the potential of application RM2 kit for imaging GRP-R overexpressed tumors, such as prostate and breast cancer. The Authors described both chemical and biological characteristics of synthesized radiobioconjugate with using pharmacopeia recommended techniques. Disadvantage of in vitro studies is using only one cell line (PC-3).
In general, manuscript is clearly described and contains all the relevant research elements in this field, along with their interpretation. Several major revisions may improve the manuscript before further processing and publication in Pharmaceutics.
- In section “Optimization of 68Ga radiolabeling” please specify what was 68Ga activity used for labeling.
Thank you for your remark, yes 68Ga-activity is an important factor. For all experiments we have used a broad range of activity from 350 to 900MBq of generator eluate. The activity has not impact on radiolabeling yield and radiochemical purity. We have added this information in the manuscript in the material and method.
- In section “Optimization of 68Ga radiolabeling” please explain why did you chose this specific volumes for testing (taking into account that 1.1 mL in each of them is 68Ga).
Thank you for your comment. As stated by the reviewer the gallium eluate has a specific volume of 1.1mL. Therefore, we were interested in testing whether the reactional volume influences the radiolabeling yield. We tested volumes that could be convenient for radiopharmacies (2mL or 4mL buffer). Also, we tested a smaller volume of 1.4mL of buffer to increase the volumic activity and see if purity and/or radiolabeling yield were increased. This volume of 1.4mLis easier to prepare using syringes than 1.5mL. We have added to the manuscript this explanation. Again, thank you for your comment
- In section “Determination of specific cellular internalization and membrane-bound fraction of [68Ga]Ga-RM2” you mentioned, that for membrane-bound fraction removal, acetate buffer with pH 5.0 was used. Usually, for this step, glycine or any other buffer with pH around 2.5-3.0 due to its more acidic properties. I am not sure, that quite significant difference in pH has no influence for effectiveness of membrane-bound conjugates removal. Please comment. Thank you for addressing this important comment. In our previous study on a radiolabeled peptide targeting the neuropeptide-Y1 receptor (Chastel et al “Design synthesis and biological evaluation of a multifunctional neuropeptide-Y conjugate for selective nuclear delivery of radiolanthanides” EJNMMI Res 2020), we have compared (unpublished) the membrane bound fraction obtained by acetate buffer pH 5 described in this work to that of 50mM glycine, 100mM NaCl, pH 3. We found no differences. Therefore, we can reasonably use acetate buffer pH5 to accurately determine the membrane bound fraction.
- In section RESULTS “Optimization of RM2 kits radiolabeling with 68Ga” please add iTLC and/or HPLC confirmation of RCP results. Thank you for your remark. We have added TLC in the paragraph “influence of heating time” as it is the parameter which influence the most the radiolabeling. For other items “reaction volume” and “temperature”, the differences are tiny and TLC or HPLC would not be able to illustrate these differences.
- In section “Comparison of the kit formulation to the module production route” please detail what is additional purification step in module synthesis. Thank you for your remark. We have added that the purification step by module route was C18 sep-pack
- Please make spaces between digit and unit (e.g. Materials and Methods; Description of RM2 kits, 5mg, etc.). Thank you for your remark regarding typo errors. There have been corrected
Summary:
Studies described in this manuscript were performed, described and interpreted carefully and all of the results were precisely discussed. Of course, using only one cell line is not the most appropriate in case of in vitro studies and might be improved. Worth attention is fact, that Authors followed European Pharmacopeia criteria for evaluation of radiobioconjugate application potential. After adding iTLC or HPLC confirmation of RCP results manuscript will be significantly improved.
We agree with the reviewer that one cell line would not be the most appropriate but when checking the literature, most developments of GRP-based radiopharmaceuticals have been performed on the PC-3 cell only. Thank you
Reviewer 6 Report
In my opinion the revised manuscript is ready for publication in Pharmaceutics.
Author Response
In my opinion the revised manuscript is ready for publication in Pharmaceutics.
Thank you very much for your positive comment
Round 2
Reviewer 2 Report
The manuscript entitled "68Ga-radiolabeling and pharmacological characterization of a kit-based formulation of the Gastrin-Releasing Peptide Receptor (GRP-R) antagonist RM2 for convenient preparation of [68Ga]Ga-RM2" submitted by Chastel A et al describes a kit based formulation of GRP-R antogonist RM2 with 68Ga radiolabeling. Please see comments below:
1) The abstract has typos, please correct and proofread them.
Rev2: [68Ga]Ga-RM2 is a potent Gastrin-Releasing Peptide-receptor (GRP-R) antagonist for imaging prostate cancer and breast cancer, currently under clinical evaluation in several specialized centers around the
word. Please correct here, does it mean word or world?
Please describe or reference what RM2 is ? Thank you for your remark. We have added a new figure showing the structure of the RM2 peptide
2) For optimization of 68Ga radiolabeling, when does RM2 was added and how the labeling works? Does 68Ga coordinates with RM2? Optimization procedure was not clear. What does "generator was eluted 4h before radiolabeling or not" What does it tells? The methodology we used is very common for radiolabeling with radiometals (see our review Morgat et al, gallium-68: chemistry and radiolabeled peptides exploring different oncogenic pathways. 2013). The methodology used to radiolabel the kits is described in details in the manuscript. Briefly, the acetate buffer was first prepare and added to the kit. Next the generator is eluted in the kit dissolved in acetate buffer. Regarding the generator, authors would provide to the reviewer the following information. A gallium generator is composed of a mother radionuclide 68Ge (adsorbed into a metallic column) which decays by electronic capture in pure 68Ga. Physical half-life of 68Ge is 271 days vs 68 min for 68Ga. Finally 68Ga decays in 68Zn, a stable metal. Therefore, the gallium generator contains metals which can interfere with the radiolabeling and can be cleaned by an elution. This is the reason why we tested the influence of a 4h-pre elution and direct labelling without 4h pre-elution. 4h is the best compromise between 68Ga-activity available and accumulation of metal and is largely used in centers equipped with 68Ga. A comprehensive review by Velikyan et al 2015 explains in details all the rational of this approach. For more clarity we have rephrase according: “Influence of 4h pre-elution was also tested”. Thank you
3) How does the author confirmed the radiolabeled 68Ga-RM2 was obtained? Thank you for your remark. We have confirmed that 68Ga-RM2 was obtained by injected natGa-RM2 into HPLC. Both 68Ga-RM2 and natGa-RM2 have indeed the same retention time. This has been added to the manuscript: [natGa]Ga-RM2 was used to confirm the chemical identity
Rev2: Does the authors confirmed the kit radiolabeling without ascorbic acid as scavenger and 50 mg of trehalose? Please include the control reactions and discuss the differences.
4) For specific cellular internalization, does the authors determine uptake of 68Ga-RM2 per mg or microgram of protein after decay and determine the protein concentration? Please provide pictures for internalization and clear procedure. The internalization is expressed in % of specific binding according to total binding. We do not provide uptake per milligram of protein because decay (> 10h) is needed before measuring protein concentration. However, PC-3 cells grow fast and the > 10h-waiting needed for decay would greatly modify the protein concentration. Therefore, we provide uptake expressed in % of specific according to total binding. The procedure for internalization is fully described in the paragraph “determination of specific cellular internalization and membrane bound fraction of [natGa]Ga-RM2. Figure 5B showed the intracellular behavior of [68Ga]Ga-RM2 on PC3 cells
Rev 2: The remaining lysates protein concentration could have been measured after complete decay of activity using a BCA assay. This should give the internalized protein concentration and need to be reported.
5) Please provide size exclusion column chromatography for RM2 at 0 min, 2 months and show SEC chromatograms for the manuscript? Thank you for your proposal. SEC is not usual for radiopharmaceutical development. Moreover, in the seminal paper of Mansi and co-workers (Mansi et al Eur J Nucl Med Mol Imaging 2011), who developed RM2, authors have used UV-radioHPLC not SEC. Therefore, we are not very comfortable in using SEC for these reasons.
Thank you, including the structure makes it clear as a peptide and 68Ga complexing.
6) In fig 1B, what are 1621.8 Da and 801.4 Da fragments refers to? Please analyze and discuss. Thank you for your remark, we were also interested in knowing what these fragments are. The 1621.8Da results in the loss of 17Da fragment from the parent peptide RM2 (1638.8Da). We hypothesized that the loss of a water molecule would be an explanation. However, the loss of water would be 18Da and not 17Da, but the loss of hydroxy is also not correct because the position of hydroxy loss would have a hydrogen in its place (CO2H to CHO) and would be a loss of 16Da not 17Da. Loss of water is more likely than the reduction of an acid to aldehyde in peptide synthesis due to the use of dehydrating reagents used in the coupling steps. The only mass we could find was hydrolysis of the C-terminal amide to an acid which then forms a 7-membered ring with the hydroxyl of the Statine and then the mass fits. 17Da would also be the loss of ammonia NH3, which could indicate glutarimide formation. As we are not able to present experimental data supporting these hypothesis we suggest to state that no obvious ideas and further work would be needed to identify what these fragments are. This is the reason why we do not give more information within the manuscript. Thank you
7) Please include SEC for 68Ga-RM2 at 100 oc and 120 oc and test their thermal stability? Please see our comment regarding SEC at point 5.
8) For figure 2, please include uv trace and coinjection of cold RM2 for identification of RM2. If RM2 kit was a mixture of components mentioned in page 3, where does Ga68 binds? Thank you for your remark. Ga3+ ion is a metal which require coordination chemistry. Therefore, the only position where Ga3+ is bound is within the DOTA macrocycle (part of the RM2 peptide). As suggested we have modified the figure by adding the UV trace of [natGa]Ga-RM2
9) For fig 3c, how did the authors determine extracellular activity? Regarding this point, the methodology is fully described in the paragraph “determination of cellular efflux of [68Ga]Ga-RM2. Briefly, cells were incubated with the radiopharmaceutical and at selected time points, externalized radioactivity in the culture medium was collected and measured in a gamma counter.
It would be ideal to address the comments for this manuscript as moderate revisions.
Author Response
[68Ga]Ga-RM2 is a potent Gastrin-Releasing Peptide-receptor (GRP-R) antagonist for imaging prostate cancer and breast cancer, currently under clinical evaluation in several specialized centers around the
word. Please correct here, does it mean word or world?
We would express our deep apologizes for this mistake. This has been corrected. Indeed the missing word was "world".
Rev2: Does the authors confirmed the kit radiolabeling without ascorbic acid as scavenger and 50 mg of trehalose? Please include the control reactions and discuss the differences.
Thank you for your remark which address an important point. First, in our hands radiolabeling RM2 with 68Ga without ascorbic acid yields lower radiochemical purities as it is sensitive to radiolysis. Regarding trehalose, kits without trehalose have not met criteria for lyophilization. Moreover, 68Ga-RM2 in module is produced without trehalose meaning that trehalose does not interfere with 68Ga-radiolabeling. We have added this discussion in the manuscript.
Rev 2: The remaining lysates protein concentration could have been measured after complete decay of activity using a BCA assay. This should give the internalized protein concentration and need to be reported.
Thank you for your remark. Between the two rounds of review, we measured the protein content. Therefore, we are very pleased to add in the manuscript the protein concentration being internalized and membrane bound. The amount vary according to the specific activity obtained.
We would like to thank the reviewer for his comments which greatly improved the quality of the manuscript
Round 3
Reviewer 2 Report
Rev 2: The revised manuscript is still not reading well. for example, In process validation section, quality control part, line 4, it says "molar was calculated"... hope it means molar activity?
Rev2: Does the authors confirmed the kit radiolabeling without ascorbic acid as scavenger and 50 mg of trehalose? Please include the control reactions and discuss the differences.
Thank you for your remark which address an important point. First, in our hands radiolabeling RM2 with 68Ga without ascorbic acid yields lower radiochemical purities as it is sensitive to radiolysis.
Rev2: Thank you, Please report radiochemical purity and radiochemical yield without ascorbic acid as a control.
Regarding trehalose, kits without trehalose have not met criteria for lyophilization. Moreover, 68Ga-RM2 in module is produced without trehalose meaning that trehalose does not interfere with 68Ga-radiolabeling. We have added this discussion in the manuscript.
Rev 2: The remaining lysates protein concentration could have been measured after complete decay of activity using a BCA assay. This should give the internalized protein concentration and need to be reported.
Thank you for your remark. Between the two rounds of review, we measured the protein content. Therefore, we are very pleased to add in the manuscript the protein concentration being internalized and membrane bound. The amount vary according to the specific activity obtained.
Please proof read the manuscript to read it well and include control reactions.
Author Response
Rev 2: The revised manuscript is still not reading well. for example, In process validation section, quality control part, line 4, it says "molar was calculated"... hope it means molar activity?
Authors warmly thank the reviewer for his carefull reading of our work. Indeed it is molar activity. We have checked all the manuscript for typo errors. Thank you
Rev2: Thank you, Please report radiochemical purity and radiochemical yield without ascorbic acid as a control
As suggested by the reviewer, we now indicates that the radiochemical purity was below 80% in case of radiolabeling withtout ascorbic acid. Reporting yield looks not relevant as all the generator eluate is added to the kit without further purification. This is the reason why we report only radiochemical purity.